# COVID-19 Medical and Pharmacological Management in the European Countries Compared to Italy: An Overview

**DOI:** 10.3390/ijerph19074262

**Published:** 2022-04-02

**Authors:** Sergio Pandolfi, Luigi Valdenassi, Geir Bjørklund, Salvatore Chirumbolo, Roman Lysiuk, Larysa Lenchyk, Monica Daniela Doşa, Serafino Fazio

**Affiliations:** 1High School Master of Oxygen-Ozone Therapy, University of Pavia, 27100 Pavia, Italy; sergiopandolfis2@gmail.com (S.P.); luigi.valdenassi@unipv.it (L.V.); 2Council for Nutritional and Environmental Medicine (CONEM), 8610 Mo i Rana, Norway; geir@vitalpress.no; 3Department of Pharmacognosy and Botany, Danylo Halytsky Lviv National Medical University, 79010 Lviv, Ukraine; pharmacognosy.org.ua@ukr.net; 4Department of Standardization Kharkiv, National University of Pharmacy, 61002 Kharkiv, Ukraine; larysa.lenchyk@nuph.edu.ua; 5Department of Pharmacology, Faculty of Medicine, Ovidius University, 900527 Constanța, Romania; monicadanielad@yahoo.com; 6Department of Internal Medicine, University of Naples Federico II, 80138 Naples, Italy; fazio0502@gmail.com

**Keywords:** COVID-19, SARS-CoV-2 epidemiology, paracetamol, deaths rate, healthcare system

## Abstract

(1) Background: Italy accounts for more than 150,000 deaths due to the COVID-19 pandemic, leading the top rank in SARS-CoV-2-caused deceases in Europe. A survey on the different ways by which the COVID-19 pandemic emergency was managed in the foreign European countries compared to Italy is the purpose of this paper. (2) Methods: A literature search and various mathematical algorithms to approach a rank scoring scale were used to describe in detail the different approaches used by European countries to manage the COVID-19 pandemic emergency. (3) Results: The study showed that Italy stands at the bottom ranking for COVID-19 management due to its high mortality rate. Possible causes of the observed huge numbers of hospitalization and deaths were (a) the demographic composition of the European country; (b) its decentralized healthcare system organization; (c) the role of correct pharmacology in the early stages before hospitalization. Post-mortem examinations were of paramount importance to elucidate the etiopathogenesis of COVID-19 and to tailor a suitable and proper therapy in the early symptomatic stages of COVID-19, preventing hospitalization. (4) Conclusions: Factors such as the significant impact on elderly people, the public health organization prevalently state-owned and represented mainly by hospitals, and criticism of the home therapy approach toward SARS-CoV-2-infected people, may have concurred in increasing the number of COVID-19 deaths in Italy.

## 1. Introduction

The huge number of deaths reported during COVID-19 in Italy has raised great concerns about the management of the COVID-19 pandemic emergency in the European Union (EU) countries, and more generally in Europe [1,2,3,4,5,6,7,8]. In particular, it seems that Italy suffered from excess COVID-19-caused mortality when compared to other European countries [6,7,8]. Experts are still wondering if the major causes of this excess have to be attributed to the Italian Government’s policy or, rather, to geographical and environmental factors [1,2,3]. People are still asking if political interventions on the basis of medical expertise and updated scientific knowledge, were effective in dampening the epidemiological impact of cases, hospitalization and deaths on the general population. A thorough study on age distribution in the whole resident population of European countries may elucidate the level of susceptibility of citizens to the COVID-19 pandemic.

Italy is particularly vulnerable to the SARS-CoV-2 pandemic, as its demographic distribution accounts for 22.8% of elderly people, compared with the 20.3% averaged mean values from other European countries [9,10]. This makes it particularly crucial for any political intervention of public health on this frail population.

For example, one of the leading topics in the debate about COVID-19 mortality in Italy regarded the way by which SARS-CoV-2-positive subjects were treated once aware of being positive or being mildly symptomatic. The pharmacological therapy of COVID-19, i.e., how to treat the patient at the earliest once COVID-19 mild or moderate symptomatology occurs, still stands as an inconclusive matter of debate, as the Government policy about this issue shows a certain weakness in addressing the problem, and commonly used drugs to relieve early symptoms such as fever, pain and discomfort raise controversial issues [11,12,13,14,15].

The current literature assessed that elderly people are particularly deficient in their glutathione (GSH) intra-cytoplasmatic levels. SARS-CoV-2 infection inhibits GSH synthesis, and the use of paracetamol further worsens GSH bioavailability, thus exposing older adults to a marked reduction in their anti-oxidant and anti-inflammatory response [11,12,13,14,15]. This may be a cause of the reported increase in hospitalizations for COVID-19 exacerbation [13,14]. Although no direct link between the use of paracetamol and excess of COVID-19 mortality can be outlined, the increase in the hospitalization rate due to COVID-19 clinical worsening should ultimately suggest an increased risk of death due to COVID-19 exacerbation, hospital-acquired infections in frail subjects and poorly targeted anti-inflammatory therapy in the early stages of SARS-CoV-2 infection [11,12].

A missing point in the attempt to approach the most correct pharmacological therapy is post-mortem investigations on COVID-19-caused deaths, which were poorly accomplished in the political decisions of many European countries, including Italy. Insights from autoptic data should have tailored the correct pharmacological protocol to treat SARS-CoV-2-positive subjects at the earliest and to prevent their immediate hospitalization, thus dampening the overload of hospital and healthcare facilities.

Addressing patients that feel the first flu-like symptoms and that are yet unaware of being COVID-19-positive, may be crucial, as any timely therapy intervention with proper pharmaceutical protocols should significantly reduce the impact on hospitalization and hence virus circulation in the community [12,13,14].

Fever is one of the leading symptoms in the earliest manifestations of COVID-19 [16]; therefore, antipyretics such as paracetamol are common in their use. In the UK, for example, despite the pack size restrictions imposed in 1998, paracetamol abuse and intentional overdoses continue to be a matter of great concern for public health in the country, and yet, the UK selected paracetamol as the leading pharmaceutical drug to address COVID-19 at its symptomatic beginning, rather than easily affordable non-steroidal anti-inflammatory drugs (NSAIDs) [17,18,19]. Recent surveys reported the extent of paracetamol availability in Europe, as pack size restrictions associated with sales from outlets and non-pharmacy services and the frequency of inquiries to the poisons information centers (PICs) moved the presence and use of this pharmaceutical drug throughout many European countries [20].

However, as outlined in this paper, the use of paracetamol in the early stages of SARS-CoV-2 infection cannot be the only radical cause of COVID-19 exacerbation and hence possible death. Elderly people are frequently affected by chronic comorbidities, usually involving metabolism, such as obesity, hypertension or even type 2 diabetes. However, it is undoubtedly correct to envisage that treatment with only a painkiller drug (such as paracetamol) in frail subjects such as elderly people, particularly while staying at home with COVID-19 early symptoms, should lead to COVID-19 exacerbation because the pain- and fever-relief pharmaceutical drug does not target the causes underlying the progression of the disease, which is of an immuno-thrombotic nature (see below).

Further causative components participate in worsening the management of the COVID-19 emergency, yet a leading role should be adopting a pharmacological strategy in mild COVID-19 patients with only pain-relief drugs instead of COX-2 inhibitors and anti-aggregate drugs such as acetylsalicylic acid (ASA) [11,12,13,14,15]. Elderly people who are weakly treated during the first days of SARS-CoV-2 infection have a great risk of COVID-19 worsening.

Data about the use or abuse of paracetamol during the SARS-CoV-2 pandemic should give insights into which type of drugs were used in the home treatment of COVID-19 upon an early onset of symptoms. As fever is one of the leading symptoms in the initial stages of COVID-19, i.e., in mild and moderate COVID-19, particularly in the elderly [16,21,22], it is conceivable that during the pandemic, a huge supply of paracetamol was requested [23], thus enormously enhancing the burden of possible adverse effects associated with its use. Paracetamol was particularly recommended when the scientific community expressed some perplexity on the use of cyclooxygenase 2 (COX-2) inhibitors, such as NSAIDs (e.g., ibuprofen), due to their ability to increase ACE2 expression [24,25]. However, NSAIDs were recently acknowledged as being able to greatly relieve fever, headache and generalized fatigue following an anti-SARS-CoV-2 vaccination. In addition, any previously purported worsening effect on COVID-19 pathogenesis by ibuprofen and other NSAIDs was rejected [26,27,28,29].

So far, no Consensus Panel, Pharmaceutical Committee or any widely agreed protocol has been attempted in Europe to successfully treat patients with COVID-19 at the occurrence of the earliest signs while they are still at home [12]. A huge component of empiricism is probably delaying this process [30,31,32,33].

To date, COVID-19 therapy remains a concerning matter of debate, as the major bulk of suggested pharmacology is limited to a few anti-viral drugs, such as remdesivir, which actually failed to address the emergency with real, encouraging results [34]. Furthermore, monoclonal antibodies are a good solution, but treatment is performed in healthcare centers such as hospitals, so their impact on reducing hospitalization is negligible. Moreover, Italy missed the thorough use of monoclonal antibodies, investing its policy, instead, quite completely in the vaccination campaign.

On the other hand, the complex COVID-19 pathogenesis appears to be halted when the patient is diagnosed within a few days from the symptom onset and simply treated with a trivial panel of anti-inflammatory drugs, such as NSAIDs (ibuprofen, celecoxib, indomethacin, nimesulide and so forth). Therefore, we can suppose that the stringent factor in COVID-19 development is time. This might appear particularly comprehensible if SARS-CoV-2 immediately targets the endothelia-platelets cross-talk and starts an immune-thrombotic mechanism, even by triggering a damage associated molecular pattern (DAMP)-mediated immunity, the consequences of which are not necessarily early to occur, aside from some inflammatory symptoms [35,36].

Each country in Europe addressed the first emergency by treating COVID-19 in its early onset when the patient was usually at home using different strategies, i.e., by (a) ratifying a holding government’s official document to rule any physician’s own decision and managing the pandemic emergency from a centralized strategy or (b) by leaving physicians’ decisions solely on their own individual skills and expertise, exclusively based on professionals’ recommendations.

This is, briefly speaking, Italy’s weakest point in managing people, particularly elderly subjects, during the first contact with the SARS-CoV-2: the Ministry of Health decided to manage home therapy via a National Recommendation Panel with paracetamol or NSAIDs and with a wait-and-watch attitude, whereas Italian physicians were left to their personal expertise without any sound and frequent cross-talking or professional debating. This left public health with quite empirical and poorly organized approaches.

In other Mediterranean countries, for example Spain, a survey about COVID-19 management at home, carried out on 3398 people, was recently performed [37]. In their cross-sectional study, the authors reported that, at least at the beginning of the pandemic, antipyretics, hydroxychloroquine and antibiotics were the most frequently used drugs to treat patients with mild symptoms of COVID-19 while staying at their homes during the lockdown [37,38]. Actually, no real COVID-19-targeting therapy was ever appointed within the first months of the pandemic, so physicians most probably were working in the dark. However, Spanish physicians referred to a multidisciplinary point of view and proposed a therapy algorithm on 20 May 2020 for COVID-19, which excluded paracetamol, to treat patients with COVID-19 at any disease stage [38]. The Health Ministry and physician boards communicated with each other. This political choice may even explain why Spain counted fewer deaths for COVID-19 than Italy.

In Italy, the first therapy algorithms or how housed people should be recommended hospitalization were proposed one year later [39,40], even though some Italian physicians addressed the need to develop a telemedicine network to manage housed COVID-19 positive subjects besides a hospitalization request [41].

Finally, we are still wondering how a high rate of COVID-19 mortality occurred in Italy.

In this review, we report an overview of how COVID-19 affected the major European counties by elaborating on publicly available data from the European Centre for Disease Prevention and Control (https://www.ecdc.europa.eu/en/cases-2019-ncov-eueea, accessed on 7 February 2022), the Italian Ministry of Health, the UK government’s website (https://coronavirus.data.gov.uk/details/download, accessed on 16 February 2022), PubMed, Scopus and other official data warehouses, using STATA Corp LLC v 17.0 for statistics. We tried to outline the various strategies adopted by different European countries compared to Italy, which were able to properly manage the COVID-19 pandemic emergency and ensure the best pharmacological therapy for patients suffering COVID-19 early symptomatology while staying in their homes.

## 2. Facing the Pandemic Emergency in the European Union (EU)

### 2.1. The Case of Paracetamol and the Concept of Wait and Watch (Monitoring) in Italy

The pandemic in Italy was particularly concerning. To date, the number of deaths caused by COVID-19 as of 31 December 2021 amounted to 137,402 (public data from the Italian Ministry of Health) and increased to 151,962 by 16 February 2022 when we are writing this manuscript, in which Italy is, so far, considered one of the leading countries for the number of deaths in Europe, exceeding 100,000 deaths alongside the UK (158,000), France (132,000) and Germany (118,000) [42]—a circumstance that should justify the severe draconian measures settled by the Italian government to date. During the first wave of the pandemic, hospitals and healthcare centers were particularly alarmed by excessive crowding and overload, and citizens with early purported symptoms of SARS-CoV-2, particularly if elderly, were recommended to stay at home, take paracetamol if with fever and wait with a watchful attitude for any exacerbating symptoms that might occur over time.

On 30 November 2020, the Italian Ministry of Health released an official document recommending paracetamol and a “wait and watch” attitude in any circumstance of SARS-CoV-2 positivity or purported infection with mild and early symptoms [11]. A civil outcry, forwarded by citizens’ and professionals’ associations, raised great criticism of the 30 November 2020 document released by the Italian Ministry of Health, promoting actions and summons in the Senate and open outcries during the press release against the simplistic proposal of paracetamol and the “wait and watch” recommendation, included in the aforementioned document, which according to many physicians and research studies, should lead to a concerning increase in hospitalized patients and in the hospitalization time [11,12,13,14]. On 26 April 2021, the Ministry of Health slightly updated this context in a newly edited document by adding the term NSAIDs [11]. Summons, complaints and administrative public trials went ahead and finally resulted in a further revision of the 26 April 2021 document on 10 February 2022, where the term “wait and watch” was replaced with the word “monitoring”. The conclusion of this thrilling controversy did not reach any serious and definitive decision. To date, Italian physicians may be perfectly authorized, by those documents, to prescribe paracetamol in place of NSAIDs upon the early symptoms of COVID-19 and counsel patients to stay home and carefully observe the evolution of their own symptoms (wait and watch) before requesting to be hospitalized due to symptoms exacerbation. The recent document from the Ministry of Health on 10 February 2022 removed the term “wait and watch” (or watchful waiting) and replaced it with the elusive “monitoring”. However, this is only a matter of terminology; the fundamental concept remains: the patient is often alone at home with simply paracetamol as a therapeutic option, unless the physician makes an alternative decision, such as NSAIDs, but the Ministry of Health never addressed a specific protocol of what NSAIDs to be used and when in any of the aforementioned documents. In brief, the formulation of an official document from the Ministry of Health may have weakened somehow the sound expertise of physicians’ own responsibility.

Concerning issues were raised in the scientific literature about this approach, particularly regarding the effect associated with paracetamol on COVID-19 and with time [11,12,13,14], as timing is stringent even for therapeutic drugs with anti-viral activity [43,44,45], despite some controversial opinions [46,47]. However, counselling, suggesting or even recommending paracetamol might be a simple consequence of common knowledge and expanded availability.

Italy realized a huge increase in paracetamol selling in 2020, as the Italian Medicines Agency (AIFA) reported paracetamol immediately following the expenditure for anxiolytics, with a burden of more than 0.3 billion of sold pharmaceuticals/year. Actually, paracetamol is the most sold active principle in the country’s pharmacy market. In 2019, paracetamol ranked third in terms of the cost burden, reaching 11.4% of the total pharmaceutical burden with a per capita expenditure of EUR 0.0360, amounting to a total cost of EUR 188 million (6.1% of the whole financial burden) [48]. As, during the SARS-CoV-2 pandemic, people may raid pharmacies in order to stockpile the drug, the government should limit paracetamol selling to ensure the widest availability. 

However, further scientific investigations are needed to assess the role of paracetamol in allowing SARS-CoV-2 infection to worsen over time, particularly because paracetamol serves as a symptom-relief agent and not as a therapy against SARS-CoV-2-induced immuno-thrombosis [40]. Furthermore, the same concept of wait and watch may have caused exacerbation in elderly people, particularly for those with comorbidities, such as hypertension and type 2 diabetes, for which vascular microcirculation is a well-known target [49,50]. It is very hard to directly associate the excess in mortality to an incorrect COVID-19 early therapy, but it is presumable that the risk of mortality can be directly associated with the increase in exacerbation.

The Italian government preferred drawing an Official Recommendation panel on how to treat SARS-CoV-2-infected people while staying at their homes, with a panel revised on 26 April 2021 by adding the option of NSAIDs in the therapy protocol after a huge outcry from citizens’ associations, professional boards’ political motion and press releases. As indicated, the latest document, on 10 February 2022, replaced the term wait and watch with monitoring.

However, the “paracetamol” option is fully questionable, as it leaves physicians to escape any personal responsibility for the possible adverse effects related to the use of paracetamol itself, particularly in SARS-CoV-2-infected elderly people, an attitude that may be felt as frustrating by many professionals.

The tale about the role of paracetamol in increasing the risk of mortality in Italy should be associated with: (a) the highest number of elderly people compared with other European countries; (b) a lack of a close discussion between the Ministry of Health and family physicians, practitioners and physicians working in the territory, about the latter’s experience in managing COVID-19 home therapy; (c) the complete absence of a safety policy completely devoted or mainly addressed to elderly people; (d) the poorly organized service of physicians in the decentralized public health. How did other European countries manage the COVID-19 pandemic emergency in order to sensitively reduce the huge impact of infected people on hospitalization?

### 2.2. Managing the COVID-19 Pandemic Emergency in the Main European Countries: A Background

Reducing the burden of severely sick people on hospitals and intensive care units is the primary goal of any policy in Europe, as hospital overload has a dramatic impact on global healthcare activity, aside from COVID-19 [51,52]. Figure 1 shows a map of Europe (top) and the impact of Intensity Care Units (ICUs’) occupancy from COVID-19-affected patients on hospitals in the European countries, according to the publicly available ECDC data updated to the end of December 2021. While France, Germany and Italy reached an amount of 4000–5000 ICUs occupancy, Sweden only about 50, Denmark 90, Portugal 200, Netherlands 500, Belgium 800 and Spain 2000, on average (Figure 1).

Fighting the COVID-19 pandemic means evaluating the ratio of victims/cases and how the country is endowed with its medical availability. The ability to combat the pandemic and manage COVID-19 cases can be assessed by calculating the following algorithm: $$S=\left(\frac{dx\delta }{1000}\right)+\varphi =B$$
where *d* = D_c_ × 1000; D_c_ = deaths/cases; α = [(*d* × *δ*)/1000]; B = (α + *φ*); *δ* = population density/km^2^; *φ* = physicians/1000 inhabitants.

The fundamental concept underlying this algorithm is that any country with a great number of COVID-19 deaths worsens its COVID-19 managing scoring if showing the highest *φ* value, as the highest number of physicians would forecast the lowest number of COVID-19 deaths. Table 1 reports the epidemiology data of COVID-19 in the main European countries for all four different COVID-19 pandemic waves, namely the first wave (from 26 February 2020 to 30 June 2020), second wave (from 1 October 2020 to 31 January 2021), third wave (from 1 February 2021 to 31 May 2021) and fourth wave (from 1 October 2021 to 31 January 2022) and cumulative data from the beginning of the COVID-19 pandemic to 7 February 2022 (data from the WHO’s Global Health Workforce Statistics OECD for country data and from the European Centre for Disease Prevention and Control, 2022). This evaluation, despite being largely approximated, provides a certain weight to each management policy implemented to address the COVID-19 pandemic in different European countries.

Management categories allow the reader to configure a defined rank position from 0 (excellent) to 5 (insufficient), considering also whether the scoring is not severe (A) or severe (B) on the basis of its closeness with the lowest (A) or highest (B) rank. Values higher than 5 are 6 (bad), 7 (very bad), 8 (concerning) and 9 (highly concerning). The values within score 3 define the return of a normal and safe circumstance, from 4 to 5 critical and beyond 5 alarming or concerning.

Figure 2 summarizes the score evaluation of European countries in the first three COVID-19 waves (Figure 2A–C). The first pandemic wave was highly concerning for countries such as the UK, Belgium Netherlands, Germany, Spain, Italy and Hungary (Figure 2A, black); high concern remained in the second pandemic wave for the UK, Belgium and Italy (Figure 2B black), then for only the UK and Italy (Figure 2C black). Considering that from the end of 2020 throughout the whole of 2021, a sustained vaccination campaign was held by any European country, we calculated at the fourth pandemic wave how many countries shifted under the score threshold of 3 (S ≤ 3B) (safety zone) due to vaccination and the low lethality of the variant of concern, B1.1.529. Figure 2D shows that all the European countries reached safe values (Figure 2D, green), except for Italy (Figure 2D, black) (see Table 1).

The case in the UK is particularly impressive.

From the third to fourth COVID-19 pandemic waves, the S value (S means score) dropped from very bad (7B) to good (3A), within the safety zone (Table 1). In the UK (data updated to 7 February 2022), the vaccination coverage is 73.1%, which is lower than in Italy (80.1%); however, the UK rescued a safe area, while Italy did not (see Figure 2D and Table 1).

However, when a Pearson’s correlation test was performed between vaccine coverage and the rate of deaths per cases for each European country, R = 0.1268, i.e., although a positive correlation can technically be reported, the relationship between variables was weak, and *p* = 0.537065 for R^2^ = 0.0161.

Countries in the 0 rank should have an S value ≤ 1.0, which includes governments adopting pandemic politics with minimal impact on daily habits, civil rights and sociality, notably reducing the impact of sick people and deaths with an excellent health policy, whereas those European countries with S ≥ 5.0 are probably adopting a government politics that have a huge impact on daily habits, civil rights and social behaviors and a very low impact on the case and mortality epidemiological curves, despite a good or excellent endowment of health facilities and expertise. Some European nations, such as Scandinavian or Baltic countries, were continuously within the safety zone. While Sweden raised controversial opinions about the Swedish policy toward the COVID-19 pandemic, Sweden maintained quite the same number of deaths (6386) in the second pandemic wave compared to the first one (5475) despite the seven-fold increase in the number of COVID-19 cases in the second wave. Additionally, in the third pandemic wave, cases did not increase significantly (staying around 490,000), but deaths dropped greatly (2319) (Table 1). Sweden has a vaccine coverage comparable to the UK (73.4%), yet the Swedish government administered a number of doses (first doses) seven-folds lower (20.8 million) than the UK (140 million). Probably, the success of Sweden may depend greatly on its government’s policy regarding the COVID-19 emergency. The UK and Sweden seem to suggest that deaths, particularly in the most recent pandemic waves, such as the fourth one, did not directly depend on the number of cases. It is fundamental to deepen this consideration, whether dependent on an excellent healthcare system or a reduction in the SARS-CoV-2 virulence.

For example, Italy had a very high number of COVID-19 deaths (147,734) despite its top position for physicians/1000 inhabitants, so it stands at the bottom of the ranking (S = 11), whereas the United Kingdom (UK), though with a higher number of deaths, has a lower D_c_ value (D_c_ = 0.8) compared to Italy (1.30), considering that the UK has only 5 physicians/1000 inhabitants (on 67 million people) and Italy 8 physicians/1000 inhabitants (on only 59 million people), which suggests the UK has better health organization compared to Italy.

Investigating the politics of European countries with the lowest S value is paramount to envisage the best approach to address the COVID-19 pandemic. Countries with δ ≤ 50/km^2^, such as Estonia, Finland, Iceland, Latvia, Lithuania, Norway and Sweden, may have lower S values (best performance) due to their population distribution. Moreover, their relative *d* values may be much higher than countries with lower S ranks. For example, Latvia (*d* = 11.000) and Lithuania (*d =* 10.653) appear to have a lower performance than France (*d* = 6.383) or the UK (*d* = 8.083), with a deaths/cases impact comparable to that of Germany (*d* = 10.683) despite the different population distributions. Therefore, a correct evaluation should consider the rate of deaths in cases related to the different population distributions and if this rate is somehow buffered by the impact of the nation’s medical endowment (Table 1 cumulative data).

Being endowed with high-tech medical resources and skilled expertise should reduce the *d* value for each European country included in this survey. High *d* values reported for Eastern European countries, such as Bulgaria (*d* = 33.925), Hungary (*d* = 25.816), Poland (*d* = 20.643) and Romania (*d* = 25.248), may be caused by the scant impact of the medical assistance on COVID-19 cases (φ < 4.417).

A thorough overview of the management efficiency during the COVID-19 pandemic emergency demonstrates that Western European countries are the worst ones, except for Portugal and Spain.

### 2.3. Managing the COVID-19 Pandemic Emergency in the Main European Countries: Portugal and Spain

Spain (*d* = 9.172) and Portugal (*d* = 6.935) do not differ so much in the rate of COVID-19 deaths from other major European countries, such as Germany (*d* = 10.683), France (*d* = 6.383) or the UK (*d* = 8.083), yet, while Portugal is worsening its political management (cat = 3A), Spain is somewhat improving its emergency program (cat = 2B) (Figure 2).

Actually, both Iberian countries managed the COVID-19 pandemic in an excellent way. To the best of our knowledge, the Spanish government never addressed official documentation to recommend resident physicians and practitioners on how to treat SARS-CoV-2 in their homes and prevent their hospitalization. Therefore, physicians were left to their own responsibilities to approach the best therapy protocol for COVID-19 patients at home to prevent hospitalization.

Actually, a Recommendation Panel of the Working Groups from the Spanish Society of Intensive and Critical Care Medicine and Coronary Units (SEMICYUC) to manage COVID-19 severely ill patients was recently addressed, but it did not report paracetamol to people before hospitalization, only paracetamol with NSAIDs and metamizole to induce lighter levels of sedation after withdrawal of the neuro-muscular blockade [53]. Further clinical protocols for COVID-19 treatment were proposed in Spain, and those with controversial therapy were not confirmed [54,55]. The Spanish National Health System is not very different from the one existing in Italy, as it was decentralized in 2002 and, like Italy, at a regional level, fueled mainly by taxation and available to any citizen free of charge [56,57].

However, only 43% of hospitals are public in Spain, whereas in Italy, this amount reaches 80% [57]. In Italy, basic decentralized medicine is organized on a territorial basis in Local Health Units (USL, now ASL), which are, in turn, divided into districts. There are limits on the number of general practitioners who can open a clinic and work in a certain area. Group clinics are still rare in Italy, and patients register with a general practitioner or pediatrician, who, in turn, can be an employee of a USL or an ASL or be an autonomous outpatient doctor affiliated with the USL/ASL.

Usually, it is the general practitioner who prescribes the specialist visits. In Italy, due to the persistent deficit of basic medicine services, the government was forced to introduce the system of participation fees or tickets. General practitioners and pediatricians who work for the NHS usually receive a fixed amount per patient. Conversely, in the Spanish government’s laws on public health stipulated that general practitioners would work in health centers serving a specific geographic area. In health centers, both health education and the use of clinical targets have established themselves well. Each center serves a population varying between 5000 and 30,000 people.

A similar organization in “healthcare communities” can be found in Portugal.

In this European country, hospitals are divided into two basic categories: central hospitals and general district hospitals. The central hospitals are located in Lisbon, Porto and Coimbra. They provide all forms of specialized medical assistance and in many cases are linked to university institutes with the qualification of university clinics. General district hospitals, on the other hand, have all the most common specializations and provide any assistance to in- and outpatients in population basins of 250,000–300,000 units. In addition to these two categories of hospital structures, there are specialized clinics (in maternity, pediatrics, orthopedics, etc.), psychiatric hospitals and local hospitals. The decentralization in Portugal has gained an excellent performance due to the collective organization of practitioners and health professionals in caregiving centers as communities, reducing the overall impact on emergency services in the hospitals. Each individual is registered with a general practitioner and can only change doctors in special cases. To go to a specialist, a prescription from a general practitioner or a pediatrician is essential. No participation fees are due for external medical services [58].

The healthcare system organized in the Iberian countries, with respect to Italy, is provided with more organized and less decentralized (as autonomous professionals) networks of physicians and practitioners, so it is much more able to optimally reach any request with respect to the difficulty in referring to single professionals scattered in the territory and serving too many patients.

Furthermore, from a pharmacological point of view, in Italy, practitioners and family physicians are less collectively joined in decentralized communities and therefore should mainly refer to recommendations from the central authority (Ministry of Health). In the Iberian countries, due to the existence of large, peripheral communities of professionals, expert protocols are mainly shared and agreed upon within this network and are able to revise and improve any consent guideline for public health coming from the central authority. Furthermore, the private, citizen-driven participation in this service is wider in Spain (57%) and Portugal (63%) than in Italy (18%).

To comprehend the excellent medical community organization in Portugal, it can be useful to compare Portugal (10,310,000 inhabitants) with Lombardy in Italy (10,060,000 residents): on 7 February 2022, Portugal reported 20,222 deaths caused by COVID-19, whereas Lombardy reported 37,713 COVID-19-caused deaths, almost a two-fold rate, despite the similar populations and a close number of total SARS-CoV-2 cases (Lombardy = 2,213,519, Portugal = 2,915,971).

In Portugal, the territorial, basic medicine is particularly organized.

The current system consists of a network of health centers (*centro de saude*) mostly managed by the Portuguese Government; thus, they are free of charge. In these public centers, the patient does not pay participation fees, and the doctors are paid. These centers are not hospitals but a kind of poly-ambulatorial service that reduces the income in the Emergency Units to less than 5%. More and more widespread, especially in urban areas, are private clinics for outpatients where doctors usually practice part-time and are paid for the service (partly by the patient himself, partly by third parties).

Spain and Portugal offer a wider plethora of flexible health services, notably reducing the quote of patients referring quite exclusively to hospitals via the emergency units and providing a network of professionals’ communities who can largely share protocols and experience in a collective, not individual, way, which encourages the network’s development.

The different health organization in Iberian countries may be a causative factor of the indicated S values (see Table 1 and Figure 2 and Figure 3), yet they should also be evaluated alongside the impact caused by elderly patients. Frailty is one of the strongest predictors of increased hospitalization within the many health covariates. According to some authors, any progression by one single point on the frailty scale (0–5) can be associated with an additional risk of 2.1% on average [59].

Recent data showed that the median age in the global European population is rapidly increasing, reaching 43.9 years at the beginning of 2020, i.e., that at least 50% of Europeans are older than 43–44 years. In particular, while in Cyprus, the median is 37.7 years, in Italy, the median is the highest in Europe, i.e., 47.2 years [60,61]. According to more recent EUROSTAT data, the percentage of adults older than 80 years in Italy is 6.7%, 6.0% in Spain and 5.9% in Portugal [59,62].

In conclusion, Spain (9.17‱) and Portugal (6.93‱) may have reported fewer COVID-19 deaths than Italy (13.02‱), standing on a higher S rank, because of better healthcare service organization and a reduced number of ≥80-year-old patients.

Figure 3 summarizes the different S values in EU countries, updated as of 7 February 2022.

### 2.4. Managing the COVID-19 Pandemic Emergency in the Main European Countries: People Recurring to Hospitals (Emergency Units) as the Only Alternative

The overall landscape arising from these data is that in Italy, the concurrence of a large elderly population and of an excellent health system, yet maximally state-owned, increased the use of hospitals for patients aged ≥65 years, leaving those people at a high risk of COVID-19 exacerbation due to hospital overload, hospital-acquired infections, increase in stressed caregiving personnel and in the length of waiting lists for pathologies aside from COVID-19, such as cancer.

People generally frightened or even scared of the COVID-19 pandemic rarely successfully refer to their own practitioner, who has to manage more than 1500 residents alone and may even disappoint many requests during the chaotic management of the emergency and, by contrast, are prompted to refer to hospitals. This is a usual circumstance occurring in Italy, where the number of people older than 50 years is preponderant and both patients and their families prefer to turn to emergency care units in hospitals when symptoms begin to appear.

A recent paper by Sirven and Rapp evaluated the many determinants affecting hospital admission in recent years and outlined two fundamental causative factors. First, healthy aging, including preventative measures and correct therapy, along with rigorous measures of the frailty of the population upon hospital arrival, are primary hallmarks in achieving excellent outcomes [63]. Second, the dynamic of care use is important as well [64]. Sirven and Rapp reported that if previously specialized professionals visit the patients, hospitalization is particularly reduced [59]. Offering integrated solutions and intertwined, joined collaborations between different professionals for frail subjects should result in an overall reduction in hospitalization rates [59].

Life in Italy is particularly awesome in its expectancy, despite some concerns regarding politics or the growth in the gross domestic product (GDP) or the per capita debt. In Italy, life expectancy is one of the highest, exceeding 80 years, along with Sweden, France and Spain, so assessing a primacy in the wellbeing of social lives within the European Union (EU) countries, despite the lowering effects due to the pandemic [65]. This optimal perspective puts the health system in great danger once a pandemic occurs.

France exhibited almost half of Italy’s COVID-19 mortality rate (6.38‱), but only 5.9% of people were older than 80 years due to a marked difference in the migration and social integration policy over the last ten years [66,67]. In France, two different kinds of health coverage can be considered: (a) basic protection (*sécurité sociale*), which is state-owned and to which the citizen contributes by withholding taxes on her/his paycheck and the payment of certain taxes if the resident is self-employed. Depending on the situation, a person may depend on the general scheme (managed by the CNAMTS, i.e., the Caisse Nationale d’Assurance Maladie des Travailleurs Salariés) or on the agricultural scheme (managed by MSA: Mutualité Sociale du Agriculture). These organizations reimburse around 70% of the benefits. The second type of health coverage is (b) complementary protection, which is private or affiliated, i.e., the famous “mutuelles”. These organizations, depending on the proposed contract, reimburse what the state does not reimburse. Therefore, in France, wide cooperation between public and private health management is orchestrated by the “mutuelles” [68].

A further attempt to comprehend why Italy leads the top rank in COVID-19 deaths in EU countries if we exclude the UK is crucial to planning new successful strategies to address the COVID-19 pandemic.

## 3. Before Going to Hospitals: How Can Positive Patients, Residents in the European Countries, Be Treated? The Concerning Issue of Post-Mortem Data

Briefly speaking, those countries with a significant number of ≥65-year-old people with an excellent life expectancy, a main state-owned health system and self-employment management of basic medicine that is poorly structured in joined communities of professionals, show two major weaknesses, i.e., an elevated presence of frail subjects and scant respondence of the medical service on their own. Therefore, in these countries, people are frequently used to referring to hospitals more than practitioners and family physicians. These subjects are prompted to either care by themselves or accept hasty medical advice in order to prevent the stressful fear of being hospitalized [69].

In Sweden, the Swedish government addressed a wide range of different measures to reduce the spread of SARS-CoV-2, thus mitigating the socio-economic impact of COVID-19. The way in which measures were accomplished responded to highly shared concepts and recommendations currently being used in Europe, such as (a) limiting the viral spreading in the country as much as possible; (b) ensuring the availability of optimal healthcare resources; (c) reducing the impact on critical health services, people and companies; (d) providing correct information regarding public health; (e) ensuring that any right measure is taken timely.

In summary, the excess of deaths observed in Italy may come from a combined mix of impairments in the availability of healthcare services to patients, the lack of a proper therapy protocol and the high use of simple pain-relief drugs.

The Swedish Ministry of Health never reported defined recommendations about how to treat SARS-CoV-2-infected patients with early and mild symptoms, yet the trends in using paracetamol in the European Nordic countries, including Scandinavian nations, increased notably from 2000 to 2015 [70,71]. However, the generalized sale of paracetamol in Swedish supermarkets was banned due to its large-scale abuse among citizens, as occurring in other Northern European countries [72,73,74].

In Italy, paracetamol was the most sold pharmaceutical drug in 2019 and 2020. As the Italian Ministry of Health recommended paracetamol to treat early-onset COVID-19 symptomatology of still home-staying people, this non-opioid analgesic was one of the most prescribed therapies to treat COVID-19 in mild symptomatic subjects with fever and pain.

However, paracetamol is widely diffused in European countries [20,75], despite some limitations in its selling, as described before. However, its common use cannot be directly linked to the increase in COVID-19 deaths, but rather, more probably, it is linked to an increase in the rate of COVID-19 hospitalizations and also in the length of hospital stays [13,14,45].

This issue exerts a huge impact on health economics and healthcare expenditure due to the documented increase in hospitalization. Furthermore, hospitalization length increases the risk of being affected by hospital-acquired infections, which may even exacerbate a severe morbidity condition, further leading to the patient’s death, particularly if with comorbidity [76].

The best political management to be adopted should consider the age composition of the European country population, as frailty is a typical hallmark of elderly people and the pharmacological and medical weapons available to safely, affordably and straightforwardly address frail people’s health. From this perspective, a missing link to address these concerns in many European countries was the paucity in post-mortem investigations, which should enable physicians and science to know and be fully aware of the COVID-19 etiopathogenesis and therefore planning a proper therapy protocol.

### Paucity in Post-Mortem Investigations and Their Scarcity in the COVID-19 Pharmacology Approach: A Cause of Concern

The few studies on COVID-19 post-mortem (autoptic) investigations reported that COVID-19-mediated diffuse alveolar damage has immune-thrombotic and immune-thromboembolic causes, as further outlined by Bonaventura et al. [35,77,78,79,80,81]. A systematic review of 60 consecutive forensic and 42 clinical COVID-19 post-mortem cases (>100 autopsies) reported that the major cause of diffuse alveolar damage is an immuno-thrombotic and thromboembolic mechanism driven by SARS-CoV-2-mediated inflammation [77,82]. The recent evidence by Colleluori et al. on post-mortem COVID-19 cases assessed that using NSAIDs is paramount to addressing COVID-19 as early as possible, as the inflammation-driven machinery leads to lung fat embolism, causing an unfavorable prognosis of severely affected COVID-19 patients [83].

Once the major causes of COVID-19 pathogenesis were discovered to be of an immune-thrombotic and thromboembolic nature, any European Ministry of Health, in an orchestrated debate with the scientific committees holding the pandemic emergency, should plan a therapeutic panel of recommendations considering the post-mortem data from the scientific community. In this circumstance, a simple pain-relief drug, such as paracetamol, should not be recommended, as inflammation starts early upon SARS-CoV-2 infection. Moreover, if anti-coagulants solely were the possible proposal, then, for example, paracetamol might not be recommended due to its purported interaction with those pharmaceuticals [84,85,86], despite controversial opinions appearing [87,88]. It is plausible that paracetamol interacts with the endothelial-platelets cross-talk by even reducing intracellular glutathione [11,12,89]; therefore, caution should be undertaken when a protocol to treat SARS-CoV-2-positive subjects, especially if elderly, includes paracetamol as an eligible pharmaceutical.

In Spain, the first COVID-19 autopsy was performed during the early stages of the pandemic, following an autopsy program in the University Hospital Ramón y Cajal in Madrid and reported without listing any author [90]. In this study on a 54-year-old male cadaver, the pathologists found hyperplastic pneumocytes with cytopathic-like changes, prominent nucleolus, multinucleation and granular cytoplasm, an exudative stage of diffuse alveolar damage and the presence of abundant thrombi in medium and small-sized vessels, expressing the platelets’ marker CD61 [90]. This autopsy was carried out on 14 February 2020 in Spain [91]. In Italy, the more recent autoptic examination on COVID-19 cadavers reported in February 2022 on eight cases (five males, three females) showed microthrombi formation and interstitial lymphocytic infiltrates [92]. Another Italian group examined 29 autopsies on COVID-19 deaths from October 2020 to February 2021, of which 21 patients died in hospitals and 7 outside; they did not report any anatomopathological findings, simply that those cadavers were SARS-CoV-2-positive at a swab test and how much [93].

Further data are from the Sacco Hospital in Milan in a retrospective cohort study conducted on autopsies from 29 February 2020 to 30 June 2020 (92 patients total and only 75 patients with fully available lung specimens) [94]. Twenty-seven patients lacking complete clinical data died partially at home (11.1%) or in the emergency units (22.2%), aside from other hospitals (66.7%) and showed that in all autopsies thrombi had a positive correlation with positive lung CT-scans (r = 0.409, *p* = 0.004) and inflammation, evaluated as IL-6 (r = 0.362, *p* = 0.049) [94]. Autoptic data assessed that COVID-19 deaths were associated with immune-thrombotic and thromboembolic mechanisms, even at the early stages of the pandemic.

Actually, further data from COVID-19 autoptic investigations in Italy consolidated this etiopathogenetic consideration [95,96].

The evidence reported by the literature on autoptic data strongly suggests that anti-inflammatory and anti-aggregant pharmaceutical drugs, rather than simple painkillers such as paracetamol, are much more likely to counteract the etiopathogenetic progress of COVID-19, leading to severe lung immune-thrombosis and, ultimately, death. Interestingly, N-acetyl-cysteine, a chemical precursor of glutathione, attenuates both vein thrombosis and platelet activation [97]. If the detrimental role of paracetamol on glutathione availability is further confirmed, this should emphasize the warning in avoiding paracetamol as eligible therapy in the treatment of mild and early symptomatic subjects while staying at home [11,12,13].

Aside from the vaccination campaign, no sound and effective therapy policy was conducted by the European countries. The best policy should be preventing hospitalization overloads by supporting the territorial potentials of various districts’ resident physicians and practitioners in decreasing the crowding of hospital emergency units, an effort led by successfully visiting and treating patients at their homes with the correct therapy protocol. Recognizing Italy’s mistakes is a promising step on the pathway toward the truth, as suggested by some authors [3,98,99].

## 4. Conclusions

This is a scientific report disclaiming any political intention, as we intended only to describe an objective survey on COVID-19 pandemic in Europe and the different managements of the pandemic emergency held from the different European Governments. 

In an attempt to provide insights into why Italy is leading the top ranking in COVID-19 deaths in Europe, we discussed some fundamental bullet points.

First, the different versions of the Ministry of Health Recommendations, i.e., on 30 November 2020, 26 April 2021 and 10 February 2022, never mentioned the importance of data coming from autoptic evidence to approach a correct therapy protocol in the earliest stages of COVID-19 on the basis of the etiopathogenesis resulting from the autoptic data.

Second, the early treatment with simple and commonly used painkillers, as recommended by the same government institutions, did not address the illness and left SARS-CoV-2 infections to worsen in patients.

Third, a full awareness of the demographic composition of Italy should have implemented safety procedures for elderly people as a primary target.

Fourth, better management of the decentralized medical endowment should have prevented a large number of hospitalizations, reducing the rate of mortality.

The survey so far described allows us to perform a critical representation of how Western European countries managed the COVID-19 pandemic, showing that the best performance is attributed to those nations that have decentralized medical coverage highly connected in large communities, which replaced central hospitals and dampened the hospital overload. Despite Italy’s high vaccination coverage (80.1%), along with Portugal (91.4%), Denmark (81.3%), and Ireland (79.8%) Italy reached the end of two pandemic years with a record in excess mortality. In this paper, we wondered also if a causative factor might be the physicians’ organization in the country.

We exemplified the case of Spain and Portugal because in these communities, physicians are encouraged to share information, expertise and skills, thus accelerating the development of a nationally agreed upon and successful therapy protocol. Even though Western European countries, Italy included, are provided with excellent healthcare and medical endowment, the increase in life expectancy enhanced the rate of elderly individuals, weakening the population distribution, for which the Government Health Authorities had to particularly watch out for. As outlined by the National Institute of Health in Italy in its most recent report (on 10 January 2022), more than 82.79% were ≥70 years old, and more than 92.62% were ≥60 years old.

The Italian government should have better targeted elderly citizens to drastically reduce the pandemic’s impact on the population.

This was not fully accomplished, and so, more than 150,000 COVID-19 deaths are a huge burden on the historical memory of Italian people.

## Figures and Tables

**Figure 1 ijerph-19-04262-f001:**
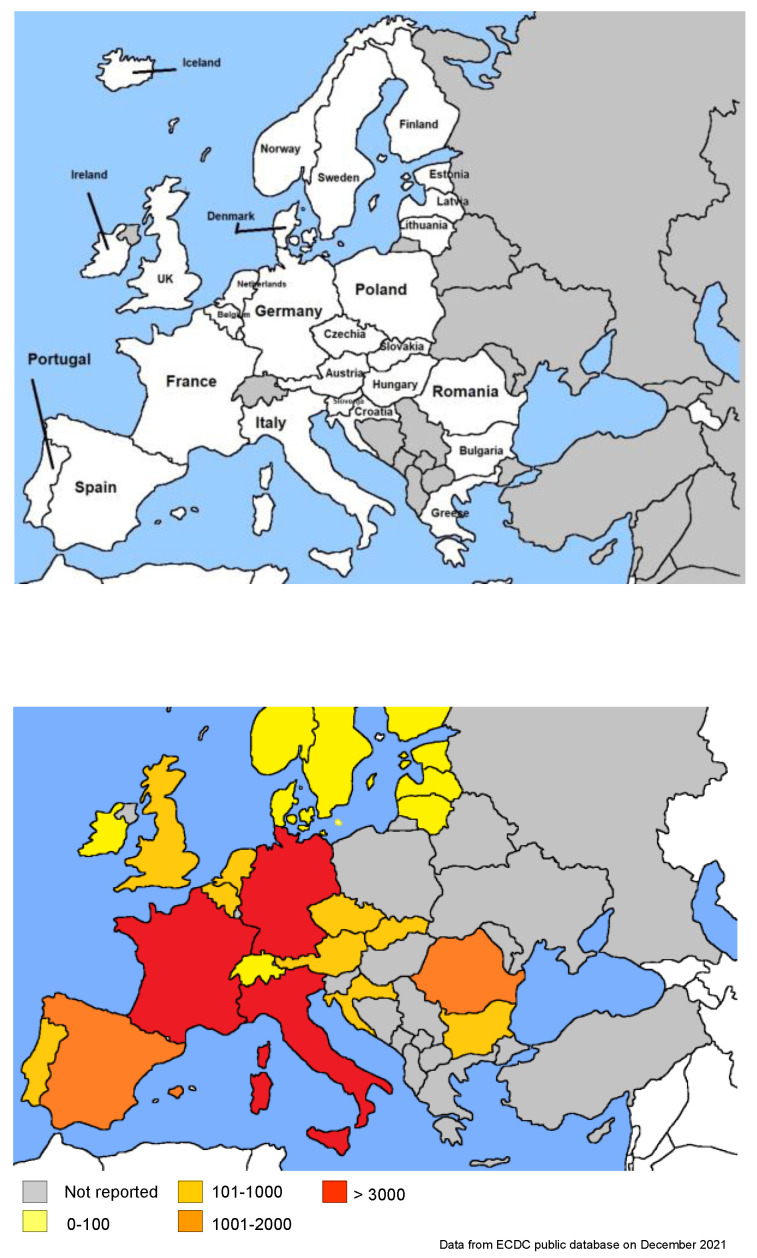
European countries considered in the study: top: map; bottom: ICUs.

**Figure 2 ijerph-19-04262-f002:**
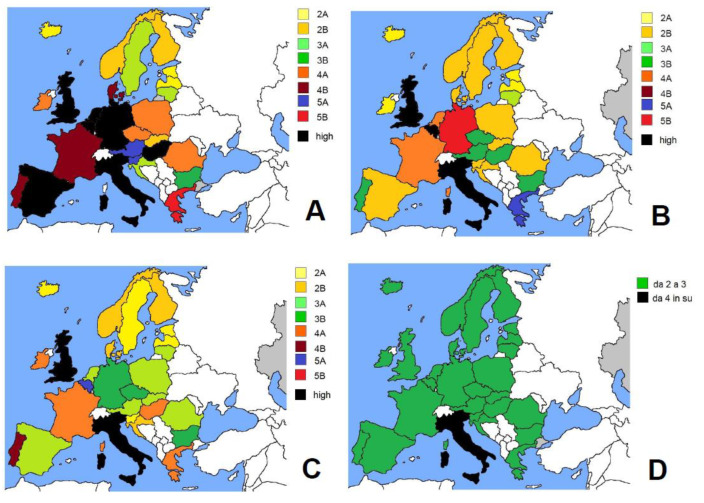
Maps of different COVID-19 waves: (**A**) 1st, (**B**) 2nd, (**C**) 3rd and (**D**) 4th. For details, see text.

**Figure 3 ijerph-19-04262-f003:**
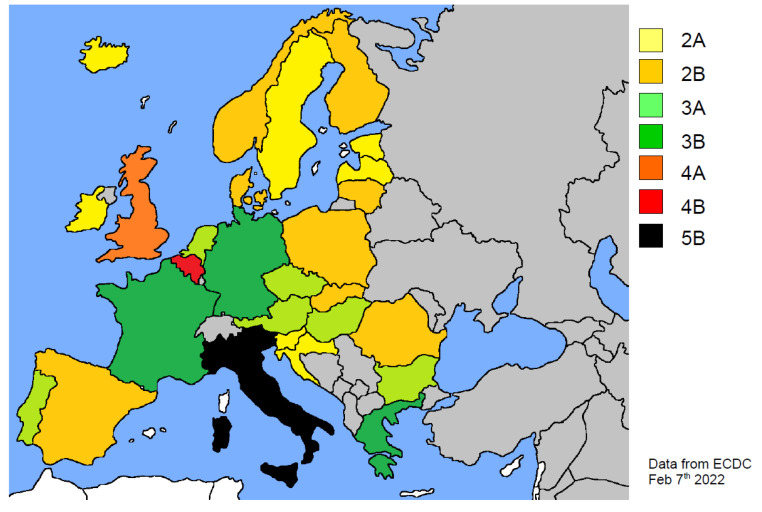
Map of European countries according to the S scoring rank (see text for details).

**Table 1 ijerph-19-04262-t001:** Overview of the COVID-19 epidemiological and demographic landscape in the main European countries during each COVID-19 wave.

EU Country	Population	Density (KM^2^)	Waves	COVID-19 Cases	Deaths	Physicians (1)	Value (2)	S CAT	
Austria	8,917,000	109	1 2 3 4	17,781 365,376 228,886 1,150,187	734 7061 2582 2830	5.211	9.710 7.316 6.440 5.479	10 7 6 5	5A 3B 3A 2B
			cumulative	2,069,496	13,719		5.934	6	3A
Belgium	11,560,000	383	1 2 3 4	61,984 584,276 350,909 1,975,730	9736 11,265 3718 3406	5.956	66.115 13.340 10.014 6.616	66 13 10 6	n.v. 6B 5A 3A
			cumulative	3,296,038	29,227		9.352	9	4B
Bulgaria	6,927,000	64	1 2 3 4	4989 197,915 199,829 439,652	223 8215 8634 12,309	4.207	7.067 6.863 6.972 5.999	7 7 7 3	3B 3B 3B 3A
			cumulative	995,436	33,770		6.378	6	3A
Croatia	4,047,000	73	1 2 3 4	2777 215,833 123,715 532,377	107 4747 2987 5187	3.000	5.812 4.605 4.762 3.711	6 5 5 4	3A 2B 2B 2A
			cumulative	983,780	14,137		4.049	4	2A
Czechia	10,700,000	139	1 2 3 4	12,017 918,710 676,653 1,348,532	347 16,197 13,507 6796	4.116	8.129 6.567 6.890 4.816	8 7 7 5	4A 3B 3B 2B
			cumulative	3,243,698	37,478		5.722	6	3A
Denmark	5,831,000	137	1 2 3 4	5797 170,474 82,755 1,303,294	214 1475 391 999	4.225	9.282 5.410 4.872 4.330	9 5 5 4	4B 2B 2B 2A
			cumulative	1,915,592	3390		4.467	5	2B
Estonia	1,331,000	31	1 2 3 4 cumulative	1985 41,142 84,985 187,072 358,087	63 357 834 672 2059	3.463	4.447 3.732 3.767 3.574 3.641	4 4 4 4 4	2A 2A 2A 2A 2A
Finland	5,531,000	18	1 2 3 4	7265 35,648 47,066 371,911	309 374 273 1140	4.640	5.405 4.829 4.744 4.695	5 5 5 5	2B 2B 2B 2B
			cumulative	514,892	2058		4.712	5	2B
France	67,390,000	119	1 2 3 4	164,787 2,633,579 2,470,210 12,127,298	29,823 44,099 33,497 14,241	6.534	8.687 8.526 8.148 6.673	9 8 8 7	4B 4A 4A 3B
			cumulative	20,758,371	132,506		7.294	7	3B
Germany	83,240,000	232	1 2 3 4	194,813 1,927,748 1,457,102 5,868,341	9256 60,629 21,571 25,383	4.300	15.322 11.596 7.734 5.303	15 11 7 5	7B 5B 3B 2B
			cumulative	11,117,599	118,766		6.778	7	3B
Greece	10,720,000	81	1 2 3 4	3409 138,482 245,349 1,284,956	192 5405 6299 8489	6.226	10.788 9.387 8.305 6.761	11 10 8 7	5B 5A 4A 3B
			cumulative	2,047,849	23,760		7.166	7	3B
Hungary	9,750,000	107	1 2 3 4	4155 341,125 436,952 730,700	585 11,759 17,209 11,215	3.408	18.473 7.096 7.622 5.050	18 7 8 5	9A 3B 4A 2B
			cumulative	1,616,846	41,741		6.170	6	3A
Iceland	366,425	3	1 2 3 4	1822 3305 604 55,834	10 19 1 13	4.142	4.158 4.159 4.147 4.143	4 4 4 4	2A 2A 2A 2A
			cumulative	73,530	49		4.144	4	2A
Ireland	5,025,898	72	1 2 3 4 cumulative	25,473 160,392 65,496 793,076 1,205,914	1736 1503 1634 887 6228	3.352	8.259 4.027 5.148 3.432 3.724	8 4 5 3 4	4A 2A 2B 1B 2A
Italy	59,550,000	206	1 2 3 4	240,256 2,238,171 1,664,789 6,310,761	34.757 52,622 37,612 15,577	8.013	37.814 12.856 12.667 8.521	37 13 13 8	n.v. 6B 6B 4A
			cumulative	11,348,701	147,734		10.695	11	5B
Latvia	1,902,000	30	1 2 3 4	1118 64,417 68,085 234,473	30 1158 1185 2156	3.302	4.107 3.841 3.824 3.577	4 4 4 4	2A 2A 2A 2A
			cumulative	450,105	4951		3.632	4	2A
Lithuania	2,795,000	43	1 2 3 4	1753 179,791 89,921 355,327	61 2765 1434 2877	5.040	6.536 5.701 5.726 5.388	6 6 6 5	3A 3A 3A 2B
			cumulative	749,616	7986		5.498	5	2B
Netherlands	17,440,000	508	1 2 3 4	50,047 857,041 671,006 2,417,928	6086 7574 3642 3095	3.707	65.483 8.196 6.464 4.357	65 8 6 4	n.v. 4A 3A 2A
			cumulative	4,892,041	21,332		5.922	6	3A
Norway	5,379,000	15	1 2 3 4	8893 49,212 62,315 616,369	249 283 226 579	4.885	5.305 4.971 4.939 4.899	5 5 5 5	2B 2B 2B 2B
			cumulative	849,436	1466		4.911	5	2B
Poland	37,950,000	124	1 2 3 4	34,393 1,421,871 1,358,898 1,979,083	1463 34,667 36,565 29,544	2.379	7.653 5.402 5.715 4.230	8 5 6 4	4A 2B 3A 2A
			cumulative	5,163,780	106,597		4.939	5	2B
Portugal	10,310,000	112	1 2 3 4 cumulative	42,171 644,974 128,577 1,570,523 2,915,971	1576 10,511 4543 1930 20222	5.312	9.497 7.137 9.269 5.450 6.089	9 7 9 5 6	4B 3B 4B 2B 3A
Romania	19,290,000	84	1 2 3 4	26,970 601,171 348,994 982,857	1651 13,510 11,977 22,984	2.981	8.123 4.869 5.864 4.945	8 5 6 5	4A 2B 3A 2B
			cumulative	2,401,821	60,642		5.102	5	2B
Slovakia	5,459,000	114	1 2 3 4	1687 428,400 332,035 789,689	28 4663 7642 5201	3.517	5.409 4.758 6.141 4.268	5 5 6 4	2B 2B 3A 2A
			cumulative	1,587,487	17,850		4.799	5	2B
Slovenia	2,100,000	103	1 2 3 4	1612 161,054 87,215 428,317	111 3353 872 1305	3.172	10.264 5.316 4.202 3.486	10 5 4 3	5A 2B 2A 1B
			cumulative	794,443	5975		3.947	4	2A
Spain	47,350,000	91	1 2 3 4	258,900 2,067,721 817,161 5,299,211	29,768 31,656 15,216 8744	4.030	14.493 5.423 5.724 4.180	14 5 6 4	7A 2B 3A 2A
			cumulative	10,271,197	94,204		4.865	5	2B
Sweden	10,350,000	25	1 2 3 4	67,866 481,077 497,919 1,030,254	5475 6386 2319 1370	4.331	6.348 4.662 4.447 4.364	6 5 4 4	3A 2B 2A 2A
			cumulative	2,287,785	16,143		4.507	4	2A
UK	67,220,000	281	1 2 3 4	312,654 3,356,998 651,556 9,675,268	56,199 72,664 21,564 19,137	5.823	56.332 11.905 15.123 6.379	56 12 15 6	n.v. 6A 7B 3A
			cumulative	17,866,632	158,000		8.083	8	4A

(^1^) on 1000 inhabitants; (^2^) A = (deaths/cases); B = (A × 1000); C = [(B × density)/1000]; D = (C + physicians). Highest score is the worst COVID-19 management. Category: 0–1 = 0; 2–3 = 1; 4–5 = 2; 6–7 = 3; 8–9 = 4; 10 and more = 5. A = is improving; B = is worsening.

## Data Availability

To consult the documents, elaborated data, data from the Italian Ministry of Health and ECDC, please contact the corresponding author or Sergio Pandolfi.

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
