# Peer review of "COVID-19 Medical and Pharmacological Management in the European Countries Compared to Italy: An Overview"

_ijerph, 2022, doi:10.3390/ijerph19074262_

Round 1
Reviewer 1 Report
The topic of this manuscript is very interesting, and I actually quite enjoyed reading through the manuscript. The covid pandemic is not over yet and this study has great scientific values if we can use an algorithm to comprehensively evaluate the efficacy of covid-19 patients management strategy for a country, which can further guide the government to adjust their healthcare system structure and pharmacology therapy guideline for patients with early covid symptomology appropriately for the future covid waves, thus reducing substantial unnecessary covid-related deaths. However, several points that the authors proposed still need further debating.
1) Line 471-478, line 276-284 & Line 427-432. The authors were trying to emphasize the official government recommendations/guidelines that advising people with mild covid symptoms to stay at home and use paracetamol if necessary is a main cause of higher covid-related deaths especially for elderly patients in Italy. Elderly patients with covid tend to have higher death tolls than general public, this is mainly because most of elderly patients have underlying health conditions which put them at an extremely higher risk of getting poor clinical outcomes i.e. longer hospital stays and higher mortality rate. The use of paracetamol might not be beneficial to the elderly patients with covid at the early stage; but it is hard to put a link between over recommendation of paracetamol for patients with early covid symptomology and higher covid-deaths in Italy.
2) Table 1: the data presented in Table are very interesting. However, calculation was based on data of covid cases, covid-related deaths from different countries up to 7th Feb 22. It is possible to list S values and CATs by different waves of the pandemic for each country? so that we can compare S value and CAT at different waves of the pandemic.
3) Table 1: CAT A is improving and B is worsening; how to define improving or worsening is unclear throughout the manuscript?
4) The algorithm: S = (???/1,000)+? = B: This algorithm might be appropriate to evaluate the efficacy of covid patient management strategy before the massive vaccination among the general public. For example, during the 1st, 2nd or 3rd waves of the pandemic (before Feb 2021 around), the demographic composition of a country (whether a higher percentage of elderly people); and healthcare system structure (whether centralised or decentralised); and the appropriate pharmacology recommendation for patients with early covid symptomology; those parameters would be vital to prevent substantial covid deaths. However, the current algorithm ignores the big impact of massive vaccination among general public on covid-related deaths.
For example, the UK had the very higher covid-related deaths, even higher than Italy during the 1st, 2nd and 3rd waves of the pandemic. However, during the last winter UK had a very lower death rate even it had historically higher covid cases. This would be mostly attributed to the role of massive vaccination which prevent people developing severe symptoms when get covid. The UK NHS has consistently recommended the general public with mild covid symptoms: to stay at home and avoid further contacts, take sufficient rest and also take paracetamol or ibuprofen if they feel uncomfortable. (How to look after yourself at home if you have coronavirus (COVID-19) - NHS (www.nhs.uk)).
Although UK had a higher S value from the data until Feb 22; but when compare the S values of before massive vaccination (1st, 2nd and 3rd waves) and after massive vaccination (last winter); there would be a huge difference.
5) It has been also known that the UK has a significant higher vaccination rate among general public and elderly patients than other European countries before the last winter. The algorithm may be more suitable to compare the European countries with similar vaccination rates among general public and elderly patients before the last Winter? For example, to compare Italy vs France, Germany, Spain etc.
6) Fig 1 & Fig 2: If the authors can put the name of each EU country in the maps; that would be more informative.
7) Fig 3: Can the authors provide the unit for X & Y axis? This figure is a bit confusing and difficult to understand.
Author Response
Point by point rebuttal to the Reviewers’ comments
Reviewer 1
The topic of this manuscript is very interesting, and I actually quite enjoyed reading through the manuscript. The covid pandemic is not over yet and this study has great scientific values if we can use an algorithm to comprehensively evaluate the efficacy of covid-19 patients management strategy for a country, which can further guide the government to adjust their healthcare system structure and pharmacology therapy guideline for patients with early covid symptomology appropriately for the future covid waves, thus reducing substantial unnecessary covid-related deaths. However, several points that the authors proposed still need further debating.
1) Line 471-478, line 276-284 & Line 427-432. The authors were trying to emphasize the official government recommendations/guidelines that advising people with mild covid symptoms to stay at home and use paracetamol if necessary is a main cause of higher covid-related deaths especially for elderly patients in Italy. Elderly patients with covid tend to have higher death tolls than general public, this is mainly because most of elderly patients have underlying health conditions which put them at an extremely higher risk of getting poor clinical outcomes i.e. longer hospital stays and higher mortality rate. The use of paracetamol might not be beneficial to the elderly patients with covid at the early stage; but it is hard to put a link between over recommendation of paracetamol for patients with early covid symptomology and higher covid-deaths in Italy.
Authors’ reply: This observation is right, yet we did not state that deaths in Italy were directly and exclusively caused by paracetamol and a wait and watch recommendation held by the Italian Government. If done, we revised the manuscript accordingly to prevent misinterpretations. Anyway, elderly patients, due to their high frailty, are much more susceptible to COVID-19 worsening if therapy is not adequate, therefore, despite a direct causative relationship between the use of paracetamol and the increase in COVID-caused deaths in Italy cannot be argued, a therapeutic protocol that does not address the etiopathogenetic causes of COVID-19 but is restricted to relief pain and discomfort, may increase the risk to COVID-19 exacerbation.
2) Table 1: the data presented in Table are very interesting. However, calculation was based on data of covid cases, covid-related deaths from different countries up to 7th Feb 22. It is possible to list S values and CATs by different waves of the pandemic for each country? so that we can compare S value and CAT at different waves of the pandemic.
Authors’ reply: Table 1 was revised accordingly and supported with a summarizing Figure.
3) Table 1: CAT A is improving and B is worsening; how to define improving or worsening is unclear throughout the manuscript?
Authors’ reply: We admit that this definition may be misleading. Therefore, we revised the text as follows: Management categories make the reader able to configure a defined rank position from 0 (excellent) to 5 (insufficient), considering also if the scoring is not severe (A) or severe (B), on the basis of its closeness with the lowest (A) or the highest (B) rank.
4) The algorithm: S = (???/1,000)+? = B: This algorithm might be appropriate to evaluate the efficacy of covid patient management strategy before the massive vaccination among the general public. For example, during the 1st, 2nd or 3rd waves of the pandemic (before Feb 2021 around), the demographic composition of a country (whether a higher percentage of elderly people); and healthcare system structure (whether centralised or decentralised); and the appropriate pharmacology recommendation for patients with early covid symptomology; those parameters would be vital to prevent substantial covid deaths. However, the current algorithm ignores the big impact of massive vaccination among general public on covid-related deaths.
For example, the UK had the very higher covid-related deaths, even higher than Italy during the 1st, 2nd and 3rd waves of the pandemic. However, during the last winter UK had a very lower death rate even it had historically higher covid cases. This would be mostly attributed to the role of massive vaccination which prevent people developing severe symptoms when get covid. The UK NHS has consistently recommended the general public with mild covid symptoms: to stay at home and avoid further contacts, take sufficient rest and also take paracetamol or ibuprofen if they feel uncomfortable. (How to look after yourself at home if you have coronavirus (COVID-19) - NHS (www.nhs.uk)).
Although UK had a higher S value from the data until Feb 22; but when compare the S values of before massive vaccination (1st, 2nd and 3rd waves) and after massive vaccination (last winter); there would be a huge difference.
Authors’ reply: Many thanks. The whole comment is welcome and we revised the manuscript by adding a part regarding vaccination. The algorithm cannot introduce vaccination because its inclusion in a simple formula is cumbersome, as vaccination was not performed in the same way in the various countries, it was affected by social outcries and opinions regarding vaccine hesitant people, Government’s different policies about mandatory vaccination, the role of public health and last but not least the effect of social interactions. In the algorithm we limited to join two evaluation. A) the impact of the pandemic and B) the ability of resilience of the country’s health system (evaluated as physician enabled in the territory). It is presumable that vaccination may be included in this second part of the algorithm. Anyway, we agreed in adding a fundamental part regarding the effect of vaccination, including the case of UK.
5) It has been also known that the UK has a significant higher vaccination rate among general public and elderly patients than other European countries before the last winter. The algorithm may be more suitable to compare the European countries with similar vaccination rates among general public and elderly patients before the last Winter? For example, to compare Italy vs France, Germany, Spain etc.
Authors’ reply: The revised manuscript made clearer this point, and UK appeared completely different respect to other countries, including Italy, for its Government’s policy.
6) Fig 1 & Fig 2: If the authors can put the name of each EU country in the maps; that would be more informative.
Authors’ reply: Initially we thought to add the names in the map but we realized it resulted very clumsy and unreadable in its immediate impact. So, we decided to not add names in the map, as we were persuaded that is sufficient that people compare any map with a map with names to recall any figure to its meaningful message. Anyway, Figure 1 is equipped with a map of Europe with relative country names.
7) Fig 3: Can the authors provide the unit for X & Y axis? This figure is a bit confusing and difficult to understand.
Authors’ reply: Done
Reviewer 2 Report
The topic is very interesting, but the paper requires substantive revisions.
The paper needs a significant revision of how results are presented respect to specific research questions. Clear research questions need to be formulated and results presented accordingly.
Methods have not been outlined in the paper- which databases have been searched, search strategy etc. as well as details on how mathematical algorithms have been developed.
More references are required to back up statements in the introduction
2.1. The case of paracetamol and the concept of wait and watch (monitoring) in Italy – a timeline with key interventions described in the paragraph would help the reader
Pag. 4 lines 153-155 – when?
Pag. 5 line 215 – “The ability to contrast pandemic and manage COVID-19 cases, can be assessed by 215 formulating the following algorithm” – who developed this algorithm?
Pag. 7 “Managing COVID-19 pandemic emergency in the main EU countries. Portugal and Spain” – why did you focus on Portugal and Spain? What about the other countries?
Check grammar errors and typos:
E.g. Pag. 1 line 40; “people are” not “people is”
Author Response
Reviewer 2
The topic is very interesting, but the paper requires substantive revisions.
The paper needs a significant revision of how results are presented respect to specific research questions. Clear research questions need to be formulated and results presented accordingly.
Authors’ reply: Done
Methods have not been outlined in the paper- which databases have been searched, search strategy etc. as well as details on how mathematical algorithms have been developed.
Authors’ reply: This part was revised accordingly
More references are required to back up statements in the introduction
Authors’s reply: More references were added
2.1. The case of paracetamol and the concept of wait and watch (monitoring) in Italy – a timeline with key interventions described in the paragraph would help the reader
Authors’ rebuttal: This part was revised accordingly
Pag. 4 lines 153-155 – when?
Authors’ reply: This part was revised accordingly
Pag. 5 line 215 – “The ability to contrast pandemic and manage COVID-19 cases, can be assessed by 215 formulating the following algorithm” – who developed this algorithm?
Authors’ reply: The algorithm was developed by the Department for Standardization National University of Kharkiv (Ukraina) in May 2021 by Prof Larysa Lenchyk and by Prof Serafino Fazio University of Naples.
Pag. 7 “Managing COVID-19 pandemic emergency in the main EU countries. Portugal and Spain” – why did you focus on Portugal and Spain? What about the other countries?
Authors’ reply: The manuscript considers the major European countries involved in the COVID-19 pandemic, not only Spain and Portugal. A special focus on Spain and Portugal can be justified by the interest in elucidating the Government’s policy in similar geographical and environmental areas respect to Italy (Northern Mediterranean area) and by healthcare systems highly comparable.
Check grammar errors and typos:
E.g. Pag. 1 line 40; “people are” not “people is”
Authors’ reply: Checked.OK
Round 2
Reviewer 2 Report
Authors addressed my comments, I suggest accepting the paper.